# Neural Ideal Large Eddy Simulation: Modeling Turbulence with Neural Stochastic Differential Equations

**Anudhyan Boral**
Google Research
Mountain View, CA 94043, USA
anudhyan@google.com

**Zhong Yi Wan**
Google Research
Mountain View, CA 94043, USA
wanzy@google.com

**Leonardo Zepeda-Núñez**
Google Research
Mountain View, CA 94043, USA
lzepedanunez@google.com

**James Lottes**
Google Research
Mountain View, CA 94043, USA
jlottes@google.com

**Qing Wang**
Google Research
Mountain View, CA 94043, USA
wqing@google.com

**Yi-fan Chen**
Google Research
Mountain View, CA 94043, USA
yifanchen@google.com

**John Roberts Anderson**
Google Research
Mountain View, CA 94043, USA
janders@google.com

**Fei Sha**
Google Research
Mountain View, CA 94043, USA
fsha@google.com

## Abstract

We introduce a data-driven learning framework that assimilates two powerful ideas: ideal large eddy simulation (LES) from turbulence closure modeling and neural stochastic differential equations (SDE) for stochastic modeling. The ideal LES models the LES flow by treating each full-order trajectory as a random realization of the underlying dynamics, as such, the effect of small-scales is marginalized to obtain the deterministic evolution of the LES state. However, ideal LES is analytically intractable. In our work, we use a latent neural SDE to model the evolution of the stochastic process and an encoder-decoder pair for transforming between the latent space and the desired ideal flow field. This stands in sharp contrast to other types of neural parameterization of closure models where each trajectory is treated as a deterministic realization of the dynamics. We show the effectiveness of our approach (niLES – neural ideal LES) on two challenging chaotic dynamical systems: Kolmogorov flow at a Reynolds number of 20,000 and flow past a cylinder at Reynolds number 500. Compared to competing methods, our method can handle non-uniform geometries using unstructured meshes seamlessly. In particular, niLES leads to trajectories with more accurate statistics and enhances stability, particularly for long-horizon rollouts. (Source codes and datasets will be made publicly available.)

37th Conference on Neural Information Processing Systems (NeurIPS 2023).

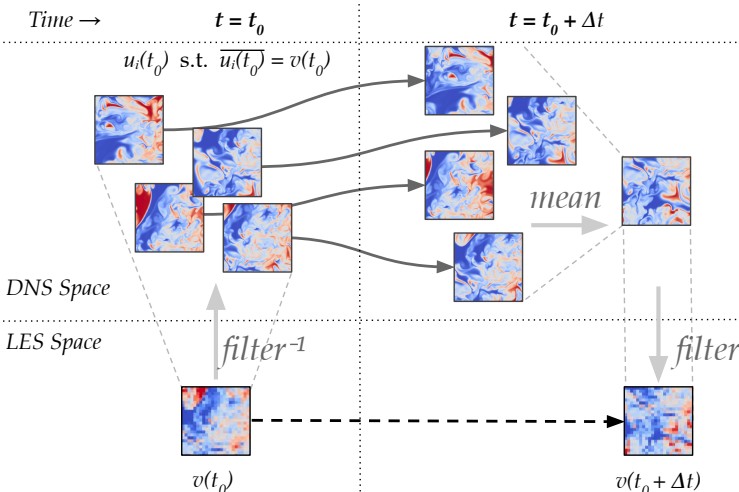

Figure 1: A cartoon of ideal LES. The LES field is lifted into the space of real turbulent fields by applying the multi-valued inverse of the filtering operator. Then the turbulent DNS fields are evolved continuously according to the N-S equations. Finally, at the end of the LES time step, the mean of the DNS fields is filtered to obtain the new LES field. Although the ideal LES is the *ideal* LES evolution, it is analytically intractable. DNS image from [54].

# 1 Introduction

Multiscale physical systems are ubiquitous and play major roles in science and engineering [5, 7, 16, 25, 57, 74, 38]. The main difficulty of simulating such systems is the need to numerically resolve strongly interacting scales that are usually order of magnitude apart. One prime example of such problems is turbulent flows, in which a fluid flow becomes chaotic under the influence of its own inertia. As such, high-fidelity simulations of such flows would require solving non-linear partial differential equations (PDEs) at very fine discretization, which is often prohibitive for downstream applications due to the high computational cost. Nonetheless, such direct numerical simulations (DNS) are regarded as gold-standard [62, 43].

For many applications where the primary interest lies in the large (spatial) scale features of the flows, solving coarse-grained PDEs is a favorable choice. However, due to the so-called back-scattering effect, the energy and the dynamics of the small-scales can have a significant influence on the behavior of the large ones [51]. Therefore, coarsening the discretization scheme *alone* results in a highly biased (and often incorrect) characterization of the large-scale dynamics. To address this issue, current approaches incorporate the interaction between the resolved and unresolved scales by employing statistical models based on the physical properties of fluids. These mathematical models are commonly known as *closure models*. Closure models in widely used approaches such as Reynolds-Averaged Navier-Stokes (RANS) and Large Eddy Simulation (LES) [68] have achieved considerable success. But they are difficult to derive for complex configurations of geometry and boundary conditions, and inherently limited in terms of accuracy [26, 67, 41].

An increasing amount of recent works have shown that machine learning (ML) has the potential to overcome many of these limitations by constructing data-driven closure models [55, 77, 47, 56]. Specifically, those ML-based models correct the aforementioned bias by comparing their resulting trajectories to coarsened DNS trajectories as ground-truth. Despite their empirical success and advantage over classical (analytical) ones, ML-based closure models suffer from several deficiencies, particularly when extrapolating beyond the training regime [9, 71, 79]. They are unstable when rolling the dynamics out to long horizons, and debugging such black box models is challenging. This raises the question: *how do we incorporate the inductive bias of the desired LES field to make a good architectural choice of the learning model?*

In this paper, we propose a new ML-based framework for designing closure models. We synergize two powerful ideas: ideal LES [50] and generative modeling via neural stochastic differential equations (NSDE) [80, 52, 44, 45]. The architecture of our closure model closely mirrors the desiderata of ideal

LES for marginalizing small-scale effects of inherent stochasticity in turbulent flows. To the best of our knowledge, our work is the first to apply probabilistic generative models for closure modeling.

Ideal LES seeks a flow field of large-scale features such that the flow is optimally consistent with an ensemble of DNS trajectories that are filtered to preserve large-scale features [50]. The optimality, in terms of minimum squared error, leads to the conditional expectation of the filtered DNS flows in the ensemble. Ideal LES stems from the observation that turbulent flows, while in principle deterministic, are of stochastic nature as small perturbations can build up exponentially fast and after a characteristic *Lyapunov* time scale the perturbations *erases* the signal from the initial condition. Thus, while deterministically capturing the trajectory of the large-scales from a single filtered DNS trajectory can be infeasible due to the chaotic behavior and loss of information through discretization, it may be possible to predict the *statistics* reliably over many possible realizations of DNS trajectories (with the same large-scale features) by marginalizing out the random effect. Fig 1 illustrates the main idea and Section 2 provides a detailed description. Unfortunately, ideal LES is not analytically tractable because both the distribution of the ensemble as well and the desired LES field are unknown.

To tackle this challenge, our main idea is to leverage i) the expressiveness of neural stochastic differential equations (NSDE) [52, 83] to model the unknown distribution of the ensemble as well as its time evolution, and ii) the power in Transformer-based encoder-decoders to map between the LES field to-be-learned and a latent space where the ensemble of the DNS flows presumably resides. The resulting closure model, which we term as neural ideal LES (niLES), incorporates the inductive bias of ideal LES by estimating the target conditional expectations via a Monte Carlo estimation of the statistics from the NSDE-learned distributions, which are themselves estimated from DNS data.

We investigate the effectiveness of niLES in modeling the Kolmogorov flow, which is a classical example of turbulent flow. We compare our framework to other methods that use neural networks to learn closure models, assuming that each trajectory is a deterministic realization. We demonstrate that niLES leads to more accurate trajectories and statistics of the dynamics, which remain stable even when rolled out to long horizons. This demonstrates the benefits of learning samples as statistics (i.e., conditional expectations) from ensembles.

## 2 Background

We provide a succinct introduction to key concepts necessary to develop our methodology: closure modeling for turbulent flows via large eddy simulation (LES) and neural stochastic different equations (NSDE). We exemplify these concepts with the Navier-Stokes (N-S) equations.

**Navier-Stokes and direct numerical simulation (DNS)**   We consider the N-S equations for incompressible fluids without external forcing. In dimensionless form, the N-S equations are:

$$\partial_t u + (u \cdot \nabla)u = -\nabla p + \nu \nabla^2 u \quad \text{with} \quad \nabla \cdot u = 0 \tag{1}$$

where $u = u(x, t)$ and $p = p(x, t)$ are the velocity and pressure of a fluid at a spatial point $x$ in the domain $\Omega \subset \mathbb{R}^d$ at time $t$; $\nu$ is the kinematic viscosity, reciprocal of the Reynolds number $Re$, which characterizes the degree of turbulence of the flow. We may eliminate pressure $p$ on the right-hand-side; e.g. by taking the divergence of the momentum equation and using the fact that velocity is divergence-free. Hence, we rewrite the N-S equations compactly as

$$\partial_t u = \mathcal{R}^{\text{NS}}(u; \nu), \tag{2}$$

We impose boundary conditions on $\partial\Omega$ and initial conditions $u(x, 0) = u_0(x)$, $x \in \Omega$ in N-S equations throughout the manuscript. We sometimes omit the implicit $\nu$ on the right hand side. To solve numerically, DNS will discretize the equation on a fine grid $\mathcal{G}$, such that all the scales are adequately resolved. It is important to note that as $\nu$ becomes small, the inertial effects becomes dominant, thus requiring a refinement of the grid (and time-step due to the Courant–Friedrichs–Lewy (CFL) condition [21]). This rapidly increases the computational cost [62].

**LES and closure modeling**   For many applications where the primary interests are large-scale features, the LES methodology balances computational cost and accuracy [76, 72]. It uses a coarse grid $\overline{\mathcal{G}}$ which has much fewer degrees-of-freedom than the fine grid $\mathcal{G}$. To represent the dynamics with respect to $\overline{\mathcal{G}}$, we define a filtering operator $\overline{(\cdot)} : \mathbb{R}^{|\mathcal{G}| \times d} \mapsto \mathbb{R}^{|\overline{\mathcal{G}}| \times d}$ which is commonly implemented

using low-pass (in spatial frequencies) filters [12]. Applying the filter to Eq. (2), we have

$$\partial_t \overline{u} = \mathcal{R}_c^{\mathrm{NS}}(\overline{u}; \nu) + \mathcal{R}^{\mathrm{closure}}(\overline{u}, u) \tag{3}$$

where $\overline{u}$ is the LES field, $\mathcal{R}_c^{\mathrm{NS}}$ has the same form as $\mathcal{R}^{\mathrm{NS}}$ with $\overline{u}$ as input, $u$ is the DNS field in Eq. (2), and $\mathcal{R}^{\mathrm{closure}}(\overline{u}, u) : \mathbb{R}^{|\overline{\mathcal{G}}| \times d} \times \mathbb{R}^{|\mathcal{G}| \times d} \mapsto \mathbb{R}^{|\mathcal{G}| \times d} = \nabla \cdot (\overline{u}\,\overline{u} - \overline{uu})$ is the closure term. It represents the collective effect of unresolved subgrid scales, which are smaller than the resolved scales in $\overline{\mathcal{G}}$.

However, as $u$ is unknown to the LES solver, the closure term needs to be approximated by functions of $\overline{u}$. How to model such terms has been the subject of a large amount of literature (see Section 5). Traditionally, those models are mathematical ones; deriving and analyzing them is highly challenging for complex cases and entails understanding the physics of the fluids.

**Learning-based closure modeling**   One emerging trend is to leverage machine learning (ML) tools to learn a data-driven closure model [47, 77] to parameterize the closure term,

$$\mathcal{R}^{\mathrm{closure}}(\overline{u}, u) \approx \mathcal{M}(\overline{u}; \theta) \tag{4}$$

where $\theta$ is the parameter of the learning model (often, a neural network). With a DNS field as a ground-truth, the goal is to adjust the $\theta$ such that the approximating LES field

$$\partial_t \tilde{u} = \mathcal{R}_c^{\mathrm{NS}}(\tilde{u}; \nu) + \mathcal{M}(\tilde{u}; \theta) \tag{5}$$

matches the filtered DNS field $\overline{u}$. This is often achieved through the empirical risk minimization framework in ML:

$$\theta^* = \arg\min_{\theta} \sum_i \|\tilde{u}_i - \overline{u}_i\|_2^2 \tag{6}$$

where $i$ indexes the trajectories in the training dataset, each being a DNS field from a simulation run with a different condition.

Despite their success, learning-based models have also their own drawbacks. Among them, *how to choose the learning architecture for parameterizing $\tilde{u}$* is more of an art than a science. Our work aims to shed light on this question by advocating designing the architecture to incorporate the inductive bias of designed LES fields. In what follows, we describe ideal LES, which motivates our work. To give a preview, our probabilistic ML framing matches very well the formulation of ideal LES in extracting statistics from turbulent flows of inherent stochastic nature.

**Ideal LES**   It has long been observed that while chaotic systems, such as turbulent flows, can be in principle deterministic, they are stochastic in nature due to the fast growth (of errors) with respect to even small perturbation. This has led to many seminal works that treat the effect of small scales stochastically [68]. Thus, instead of viewing each DNS field as a deterministic and distinctive realization, one should consider an ensemble of DNS fields. Furthermore, since the filtering operator $\overline{(\cdot)}$ is fundamentally lossy [75], filtering multiple DNS fields could result in the same LES state. Ideal LES identifies an evolution of LES field such that the dynamics is consistent with the dynamics of its corresponding (many) DNS fields. Formally, let the initial distribution $\pi_{t_0}(u)$ over the (unfiltered) turbulent fields to be fixed but unknown. By evolving forward the initial distribution according to Eq. (2), we obtain the stochastic process $\pi_t(u)$.

The evolution of the ideal LES field $v$ is obtained from the time derivatives of the set of unfiltered turbulent fields whose large scale features are the same as $v$ [50]:

$$\frac{\partial v}{\partial t} = \mathbb{E}_{\pi_t} \left[ \left. \overline{\frac{\partial u}{\partial t}} \;\right|\; \overline{u} = v \right] \tag{7}$$

Fig 1 illustrates the conceptual framing of the ideal LES. We can gain additional intuition by observing that the field $v$ also attains the minimum mean squared error, matching its velocity $\partial v / \partial t$ to that of the filtered field $\partial \overline{u} / \partial t$.

It is difficult to obtain the set $\{u \mid \overline{u} = v\}$ which is required to compute the velocity field (and infer $v$). Thus, despite its conceptual appeal, ideal LES is analytic intractable. We will show how to derive a data-driven closure model inspired by the ideal LES, using the tool of NSDE described below.

**NSDE** NSDE extends the classical stochastic differential equations by using neural-network parameterized drift and diffusion terms [45, 80, 52]. It has been widely used as a data-driven model for stochastic dynamical systems. Concretely, let time $t \in [0, 1]$, $Z_t$ the latent state and $X_t$ the observed variable. NSDE defines the following generative process of data via a latent Markov process:

$$Z_0 \sim p_0(\cdot), \ p(Z_t) \sim dZ_t = h_\theta(Z_t, t)dt + g_\theta(Z_t, t) \circ dW_t, \ X_t \sim p(X_t|Z_t) \tag{8}$$

where $p_0(\cdot)$ is the distribution for the initial state. $W_t$ is the Wiener process and $\circ$ denotes the Stratonovich stochastic integral. The Markov process $\{Z_t\}_{t \in [0,1]}$ provides a probabilistic prior for the dynamics, to be inferred from observations $\{X_t\}_{t \in [0,1]}$. Note that the observation model $p(X_t|Z_t)$ only depends on the state at time $t$. $h(\cdot)$ and $g(\cdot)$ are the drift and diffusion terms, expressed by two neural networks with parameters $\theta$ and $\phi$.

Given observation data $x = \{x_t\}$, learning $\theta$ and $\phi$ is achieved by maximizing the Evidence Lower Bound (ELBO), under a variational posterior distribution which is also an SDE [52]

$$q(Z_t|x) \sim dZ_t = h_\phi(Z_t, t, x)dt + g_\theta(Z_t, t) \circ dW_t \tag{9}$$

Note that the variational posterior has the same diffusion as the prior. This is required to ensure a finite ELBO, which is given by

$$\mathcal{L} = \mathbb{E}_q \left\{ \int_0^1 \log(x_t|z_t)dt - \int_0^1 \frac{1}{2} \left( \frac{h_\phi(Z_t, t) - h_\theta(Z_t, t, x)}{g_\theta(Z_t, t)} \right)^2 dt \right\} \tag{10}$$

Note that both the original SDE parameters $\theta$ and $\phi$ and the variational parameter $\psi$ are jointly optimized to maximize $\mathcal{L}$. For a detailed exposition of the subject, please refer to [45]. In the next section, we will describe how NSDE is used to characterize the stochastic turbulent flow fields.

## 3   Methodology

We propose a neural SDE based closure model that implements ideal LES. The generative latent dynamical process in neural SDE provides a natural setting for modeling the unknown distribution of the DNS flow ensemble that is crucial for ideal LES to reproduce long-term statistics.

**Setup** We are given a set of filtered DNS trajectories $\{\bar{u}_i\}$. Each $\bar{u}_i$ is a sequence of "snapshots" indexed by time $t$ and spans a temporal interval $\mathcal{T}_i$ over the domain $\Omega$. We use VARIABLE$(t)$ to denote the time $t$ snapshot of the VARIABLE (such as $\bar{u}_i(t)$ or $v(t)$). Those trajectories are treated as "ground-truth" and we would like to derive a data-driven closure model $\mathcal{M}(v; \Theta)$ in the form of Eq. (3) to evolve the LES state $v$

$$\partial_t v = \mathcal{R}_c^{\text{NS}}(v) + \mathcal{M}(v), \tag{11}$$

where we have dropped the model's parameters $\Theta$ for notation simplicity. Our goal is to identify the optimal $\mathcal{M}(v)$ such that the trajectories of Eq. (11) have the same long-term statistics as Eq. (2). Following ideal LES, we would render Eq. (11) equivalent to Eq. (7) by implementing the ideal $\mathcal{M}(v)$ as

$$\mathcal{M}(v(t)) = \mathbb{E}_{\pi_t} \left[ \overline{\partial_t u} \mid \bar{u} = v(t) \right] - \mathcal{R}_c^{\text{NS}}(v(t)). \tag{12}$$

Let $\bar{p}_t(u; v(t))$ denote the density of $\pi_t(u)$ restricted to the set $\{u|\bar{u} = v(t)\}$:

$$\bar{p}_t(u; v(t)) \propto \delta(\bar{u} = v(t))\pi_t(u). \tag{13}$$

We can thus rewrite the closure model as

$$\mathcal{M}(v(t)) = \int \left[ \overline{\partial_t u} - \mathcal{R}_c^{\text{NS}}(v(t)) \right] \bar{p}(u; v(t)) \, du = \int f(u; v(t))\bar{p}(u; v(t)) \, du \tag{14}$$

where $f(u)$ denotes the fluctuation exerted on the large scales $v$ if the *true* DNS trajectory was $u$. This shows that the ideal $\mathcal{M}(v(t))$ should compute the mean effect of the small-scale fluctuations $f(u)$ by integrating over all possible DNS trajectories $u$ that are consistent with the large scales of $v(t)$. However, just as ideal LES is not analytically tractable, so is this ideal closure model. Specifically, while the term $\mathcal{R}_c^{\text{NS}}(v(t))$ can be easily computed using a numerical solver on the coarse grid, the remaining terms are not easily computed. In particular, $\overline{\partial_t u}$ would require a DNS solver, thus defeating the purpose of seeking a closure model. An approximation to Eq. (14) is needed.

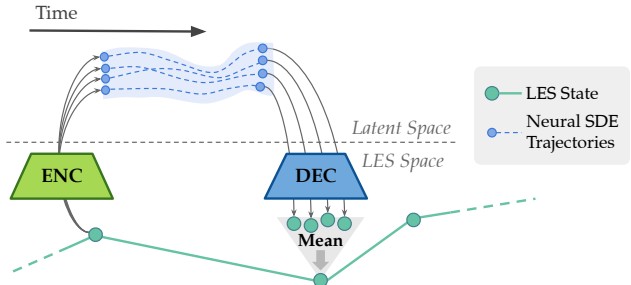

Figure 2: Schematic of our modeling approach motivated from ideal LES (cf. Figure 1 for structural correspondence). The evolution in the low-dimensional latent space follows trajectories of a data-driven Neural SDE, mirroring the fine temporal resolution of the DNS trajectories. The final states of the latent state at the next time step are decoded into the LES space and averaged over to obtain the mean correction due to small-scale fluctuations.

## 3.1 Main idea

Consider a trajectory segment from $t_0$ to $t_1$ where we are told only $\bar{u}(t_0)$ and $\bar{u}(t_1)$, *how can we discover all valid DNS trajectories between these two times to compute the closure model without incurring the cost of actually computing DNS?* Our main idea is to use a probabilistic generative model to generate a representation of those trajectories at a fraction of the cost. One can certainly learn from a corpus of DNS trajectories but the corpus is still too costly to acquire. Instead, we leverage the observation that we do not need the *full* details of the DNS trajectories in order to approximate $M(v(t))$ well — the expectation itself is a low-pass filtering operation. Thus, we proceed by constructing a (parameterized) stochastic dynamic process to emulate the *unknown* trajectories and collect the necessary statistics. As long as the constructed process is differentiable, we can optimize it end to end by matching the resulting dynamics of using the closure model to the ground-truth.

We sketch the main components below. The stochastic process is instantiated in the latent state space of a neural SDE with an encoder whose output defines the initial state distribution controlled by the desired LES state at time $t_0$. The desired statistics, *i.e.*, the mean effect of small-scales, is computed in Monte Carlo estimation via a parameterized nonlinear mapping called *decoder*. The resulting correction by the closure model is then compared to the desired LES state at time $t_1$, driving learning signals to optimize all parameters. See Fig. 2 for an illustration with details in Appendix D.

**Encoder** The encoder defines the distribution of the initial latent state variable in the NSDE, denoted by $Z_0 \in \mathbb{R}^L$. Concretely, $\mathcal{E} : \mathbb{R}^{\overline{\mathcal{G}} \times d} \mapsto \mathbb{R}^{L+L(L+1)/2}$ maps from $v(t_0)$ in the LES space to the mean and the covariance matrix of a multidimensional Gaussian in the latent space: $\mathcal{E}(v(t_0)) = (\mu_{z_0}, \Sigma_{z_0})$. This distribution is used to sample $K$ initial latent states in $\mathbb{R}^L$: $Z_0^{(i)} \sim \mathcal{N}(\mu_{z_0}, \Sigma_{z_0})$ $(1 \leq i \leq K)$.

**Latent space evolution** The latent stochastic process evolves according to time $\tau \in [0, 1]$. This is an important distinction from the time for the LES field. Since we are interested in extracting statistics from DNS with finer temporal resolution, $\tau$ represents a faster (time-stepping) evolving process whose beginning and end map to the physical time $t_0$ and $t_1$. (In our implementation, $\Delta t = t_1 - t_0$, the time step of the coarse solver, is greater than $\Delta \tau$, the time step for the NSDE.)

The latent variable $Z_\tau$ evolves according to the Neural SDE:

$$dZ_\tau = h_\phi(Z_\tau, \tau)d\tau + g_\theta(Z_\tau, \tau) \circ dW_\tau \tag{15}$$

where $W_\tau$ is the Wiener process on the interval $[0, 1]$. We obtain trajectories $\{Z_\tau^{(i)}\}_{\tau \in [0,1]}$ sampled from the NSDE, and in particular, we obtain an ensemble of $\{Z_1^{(i)}\}_{i=1}^K$.

**Decoder** The decoder $\mathcal{D} : \mathbb{R}^L \mapsto \mathbb{R}^{\overline{\mathcal{G}} \times d}$ maps each of the $K$ ensemble members $\{Z_1^{(i)}\}_{i=1}^K$ from latent state back into the LES space. So we can compute the Monte-Carlo approximation of

**Algorithm 1 Compute** $\mathcal{M}(v(t_0))$

---

**Input: LES state** $v(t_0)$**, Closure model parameters** $\Theta$

1. Encode $v(t_0)$ to a distribution on the latent state: $(\mu_{z_0}, \sigma_{z_0}) = \mathcal{E}_{\Theta}(v(t_0))$.

2. Sample $K$ initial latent states in $\mathbb{R}^L$; $Z_0^{(i)} \sim \mathcal{N}(\mu_{z_0}, \sigma_{z_0})$ $(1 \le i \le K)$.

3. Sample trajectories $Z_\tau^{(i)}$ $(\tau \in (0,1])$ by solving Eq. (15) with initial conditions $Z_0^{(i)}$.

4. Decode the final latent states $Z_1^{(i)}$ to the LES space: $\mathbf{x}^{(i)} = \mathcal{D}_{\Theta}\left(Z_1^{(i)}\right)$ $(1 \le i \le K)$.

5. Take the empirical mean of the samples: $v' = (\frac{1}{K})\sum_{i=1}^K \mathbf{x}^{(i)}$.

**Output:** $v'/(t_1 - t_0)$.

---

$\mathcal{M}(v(t_0))(t_1 - t_0)$ (cf. Eq. (14) for the definition) as

$$\int_{t_0}^{t_1} dt \int f(u; v(t))\bar{p}(u; v(t))du \approx \int dw f_w(w; v)p_w(w) \approx \frac{1}{K}\sum_{i=1}^K \mathcal{D}(Z_1^{(i)}), \qquad (16)$$

where $w$ is the spatio-temporal lifted version of $u$, i.e., the space of DNS trajectories in space and time, and $f_w$ absorbs both $f$ in the closure Eq. (14) and the implicit conditioning in Eq. (13), and $p_w$ is a prior for the trajectories. This notational change allows us to approximate the integral directly by a Monte-Carlo approximation on the lifted variables, i.e., the distribution of trajectories (as shown in Fig. 2) which are modeled using the NSDE in Eq. (15). See Algo. 1 for the calculation steps.

We stress that while we motivate our design via mimicking DNS trajectories, $Z$ is low-dimensional and is not replicating the real DNS field $u$. However, it is possible that with enough training data, $Z$ might discover the low-dimensional solution manifold in $u$.

**Training objective**  The NSDE is differentiable. Thus, with the data-driven closure model Eq. (16), we can apply end-to-end learning to match the dynamics with the closure model to ground-truths. Concretely, let $v(t_0)$ be $\bar{u}(t_0)$ and we numerically integrate Eq. (11)

$$v(t_1) \approx v(t_0) + \int_{t_0}^{t_1} \mathcal{R}_c^{\text{NS}}(v(t))dt + \mathcal{M}(v(t_0))(t_1 - t_0) \qquad (17)$$

This gives rise to the likelihood of the (observed) data in NSDE. Specifically, following the VAE setup of [52], we have

$$-\log p(\bar{u}(t_1)|Z) = \mathcal{L}^{\text{recon}}(v(t_1), \bar{u}(t_1)) = (2\sigma^2)^{-1}\|v(t_1) - \bar{u}(t_1)\|^2 \qquad (18)$$

where $\sigma$ is a scalar posterior scale term. The training objective $\mathcal{L}(v, u) = \mathcal{L}^{\text{recon}}(v, u) + \text{KL}^{\text{NSDE}}$ includes a KL divergence term associated with the neural SDE. See Appendix D for more details on the additional term.

To endow more stability to the training, following [81] the training loss incorporates $S$ time steps $(S > 1)$ of rollouts of the LES state:

$$\mathcal{L}^{(S)}(v, \bar{u}) = \sum_{k=1}^S \mathcal{L}(v(t_0 + k\Delta t), \bar{u}(t_0 + k\Delta t)) \qquad (19)$$

For a dataset with multiple trajectories, we just sum the loss for each one of them and optimize through stochastic gradient descent.

## 3.2  Implementation details

We describe details in the Appendix G for the Transformer-based encoder and decoder and Appendix D for implementing NSDE with a Transformer parameterizing the drift and diffusion terms.

# 4  Experimental results

We showcase the advantage of our approach using two instances of a chaotic Navier-Stokes flow: 2D Kolmogorov flow, [47] at a Reynolds number of 20,000 and flow past a circular cylinder at a Reynolds number of 500.

### 4.1 Setup

**Dataset generation**    The reference data for Kolmogorov flow consists of an ensemble of trajectories generated by randomly perturbing an initial condition. The flow past cylinder dataset is generated as a single long, chaotic trajectory, split up into 14 equal segments with an initial subsegment thrown away to avoid correlation between the segments. Each trajectory is generated by solving the NS equations directly using a high-order spectral finite element discretization in space and time-stepping is done via a 3rd order backward differentiation formula (BDF3) [23], subject to the appropriate Courant–Friedrichs–Lewy (CFL) condition [21]. These DNS calculations were performed on a mesh with 2304 elements for Kolmogorov flow and 292 elements for the cylinder flow with each element having a polynomial order of 8 and 13 respectively. The DNS trajectories were then sampled to a coarse grid with order 4 and 5 respectively for the two datasets. For more details see Appendix B.

**Benchmark methods**    For all LES methods, including niLES, we use a $10\times$ larger time step than the DNS simulation. We compare our method against a classical implicit LES at polynomial order 4 using a high order spectral element solver using the filtering procedure of [28]. As reported in [14], the implicit LES for high order spectral methods is comparable to Smagorinsky subgrid-scale eddy-viscosity models. We also train an encoder-decoder deterministic NN-based closure model using a similar transformer backend as our niLES model. This follows the procedure from prior works [56, 81, 47, 24] in training deterministic NN-based closure models. Additionally, we train a deterministic encoder-processor-decoder network with a neural ODE processor [19] with strictly more parameters than the niLES model.

**Metrics**    For measuring the short-term accuracy we used root-mean-squared-error (RMSE) after unrolling the LES forward. Due to chaotic divergence, the measure becomes meaningless after about 1000 steps. For assessing the long-term ability to capture the statistics of DNS we use turbulent kinetic energy (TKE) spectra. The TKE spectra is obtained by taking the 2D Fourier transform of the velocity field, after interpolating it to a uniform grid. Then we obtain the kinetic energy in the Fourier domain and we integrate it in concentric annuli along the 2D wavenumber $k$.

### 4.2 Main Results

**Long-term turbulent statistics**    We summarize in Figs. 3 and 5. In the short-term regime, both niLES and deterministic methods achieve higher accuracy than the implicit LES, even when unrolling for hundreds of steps. Beyond this time frame, the chaotic divergence leads to exponential accumulation of pointwise errors, until the LES states are fully decorrelated with the DNS trajectory. At this point, we must resort to statistical measures to gauge the quality of the rollout.

When considering reconstruction errors from very short-term rollouts, such as when the model is unrolled only 8 times, identical to the training setup, the deterministic approaches and niLES yield similar reconstruction errors. Beyond this regime, and going up to 1000s of rollout steps, the probabilistic approach of niLES provides superior generalization ability even when measuring a deterministic reconstruction loss.

Furthermore, niLES captures long-term turbulent statistics significantly better than the other two approaches, particularly in the high wavenumber regime. The deterministic NN is not stable for long term rollouts due to energy buildup in the small scales, which eventually leads the simulation to blow up.

**Inference costs**    The cost of the inference for our method as well as the baselines are summarized in Table 1. niLES uses four SDE samples for both training and inference, and each SDE sample is resolved using 16 uniformly-spaced time steps corresponding to a single LES time step. The inference cost scales linearly with both the number of samples and the temporal resolution. However, niLES achieves much lower inference cost than the DNS solver while having similarly accurate turbulent statistics for the resolved field. The deterministic NN model has slightly lower inference cost than niLES, since it can forego the SDE solves. While the implicit LES is the fastest method and is long-term stable, it cannot capture the statistics especially in the high wavenumber regime.

**Limitations**    A drawback of our current approach stems from using transformers for the encoder-decoder phase of our model, which might result in poor scaling with increasing number of mesh

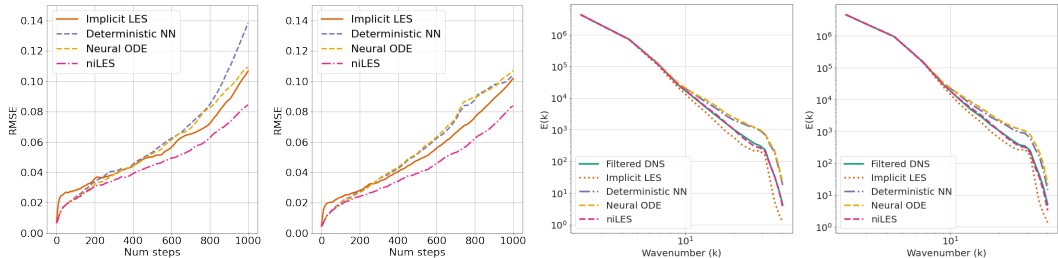

Figure 3: Root mean squared error (RMSE) over the first 1000 steps (first two columns) and the turbulent kinetic energy (TKE) spectrum $E(k)$ averaged over the first 2500 steps (right two columns) of two independent test trajectories unseen during training or validation. niLES has an improved ability to capture the long term statistics accurately compared to both implicit LES and deterministic NN. The energy buildup in the small scales (large wavenumber) in the deterministic NN model eventually leads to unstable trajectories.

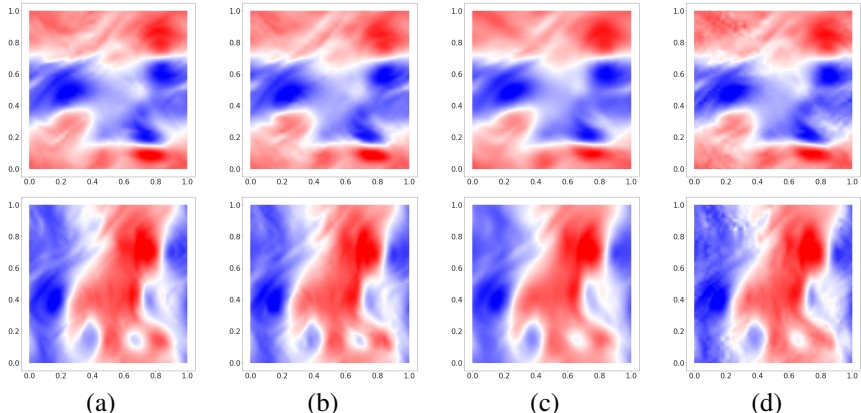

(a)                    (b)                    (c)                    (d)

Figure 4: Comparison between rollout predictions after 800 LES steps on a held-out trajectory. Velocities in the $x$ (top row) and $y$ (bottom row) directions respectively. Snapshots of filtered DNS (reference) (a), niLES (b), implicit LES (c) and deterministic NN models (d). The niLES captures several finer scale features of the flow consistent with the reference filtered DNS trajectory. The implicit LES has an overall smoothing effect and some turbulent structures are not captured. The deterministic NN LES shows artifacts which indicate instability.

elements. Alternative architectures which can still handle interacting mesh elements should be explored. Additionally, the expressibility of the latent space in which the solution manifold needs to be embedded can affect the performance of our algorithm, and requires further study.

## 5 Related work

The relevant literature on closure modeling is extensive as it is one of the most classical topics in computational methods for science and engineering [68]. In recent years, there has also been an explosion of machine learning methods for turbulence modeling [9, 25] and multi-scaled systems in general. We loosely divide the related works into four categories, placing particular emphasis on the treatment of effects caused by unresolved (typically small-scaled) variables.

**Classical turbulence methods** primarily relies on phenomenological arguments to derive an *eddy viscosity* term [48], which is added to the physical viscosity and accounts for the dissipation of energy from large to small scales. The term may be static [4], time-dependent [76, 31] or multi-scale [39, 40].

**Data-driven surrogates** often do not model the closure in an explicit way. However, by learning the dynamics directly from data at finite resolution, the effects of unresolved variables and scales are expected to be captured implicitly and embedded in the machine learning models. A variety of architectures have been explored, including ones based on multi-scaled convolutional neural networks [70, 82, 77], transformers [11], graph neural networks [73, 49] and operator learning [65].

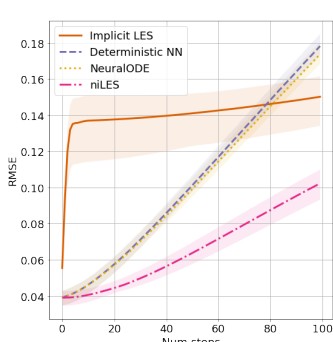

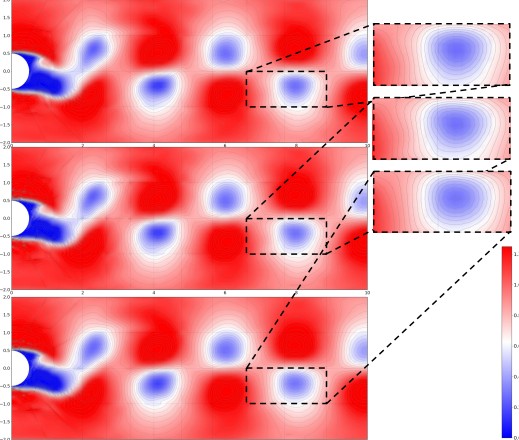

Figure 5: Average RMSE over 10 testcases on the cylinder wake dataset. Shaded regions depict the spread (minimum and maximum) among the testcases.

Figure 6: Comparison between rollout predictions of the horizontal component of velocity in the cylinder wake dataset after 100 LES steps on a held-out testcase. From top to bottom: snapshots of (a) filtered DNS (reference), (b) niLES, and (c) ILES. The niLES captures correct shapes of the vortices when compared to ILES. The latter has an excessive smoothing effect and the vortices appear more rounded.

Table 1: Inference times in wall clock seconds to evolve the state for one second of simulation time for the Kolmogorov dataset

| Method | DNS | Implicit LES | Deterministic NN | niLES (Ours) |
|---|---|---|---|---|
| Inference time [s] | 15600 | 8.85 | 11.4 | 23.8 |

**Hybrid physics-ML** contains a rich set of recent methods to combine classical numerical schemes and deep learning models [61, 6, 47, 56, 24, 81, 58, 35]. The former is expected to provide a reasonable baseline, while the latter specializes in capturing the interactions between modeled and unmodeled variables that accurately represent high-resolution data. This yields cost-effective, low-resolution methods that achieve comparable accuracy to more expensive simulations.

**Probabilistic turbulence modeling** seeks to represent small-scaled turbulence as stochastic processes [1, 32, 33, 20, 36]. Compared to their deterministic counterparts, these models are better equipped to model the backscattering of energy from small to large scales (i.e. opposite to the direction of energy flow in eddy viscosity), which is rare in occurrence but often associated with events of great practical interest and importance (e.g. rogue waves, extreme wildfires, etc.).

Our proposed method is inspired by methods in the last two categories. The closure model we seek includes the coarse solver in the loop while using a probabilistic generative model to emulate the stochastic process underlying the turbulent flow fields. This gives rise to long-term statistics that are accurate as the inexpensive neural SDE provides a good surrogate for the ensemble of the flows.

## 6  Conclusion

Due to chaotic divergence, it is infeasible to predict the state of the system with pointwise accuracy. Fortunately, in many systems of interest, the presence of an attractor allows a much lower-dimensional model to capture the essential statistics of the system. However, the time evolution of the attractor is not known, which makes building effective models challenging.

In this work we have argued that taking a probabilistic viewpoint is useful when modeling chaotic systems over long time windows. Our work has shown modeling the evolution with a neural SDE is beneficial in preserving long-term statistics and this line of thinking is likely fruitful.

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
