# A    Metrics

**Root mean square error**    The root mean square (RMSE) provides a measure of pointwise accuracy. While such a measure may not be useful for long term rollouts, due to the chaotic divergence of the system: even infinitesimal perturbations causes trajectories to diverge exponentially; it is a good proxy for short-term accuracy. Given a predicted LES state $v^{\mathrm{pred}}$ and the filtered DNS reference trajectory $\overline{u}$, we compute the root mean squared error over the domain as:

$$\mathrm{RMSE}(v^{\mathrm{pred}}, \overline{u}) = \sqrt{\int_{\Omega \subset \mathbb{R}^2} (v^{\mathrm{pred}} - \overline{u})^2 \, dx}, \tag{20}$$

where $\Omega$ is the domain of N-S system in Eq. (1). Since both $v^{\mathrm{pred}}$ and $\overline{u}$ are represented as degree $P$ polynomials on each quadrilateral element (see Appendix C), the integral is computed exactly using an appropriate quadrature rule; in this case we use the Gauss-Lobatto-Legendre (GLL) quadrature rule on each 2D quadrilateral element to compute the RMSE.

**Turbulent kinetic energy (TKE) spectrum**    The TKE spectrum provides an aggregate view of the energy content of the fluid at different scales. It is one of the key quantities used to determine the spectral accuracy of simulations in the study of turbulence [13]. In particular, the TKE spectrum captures a snapshot of the energy cascade, which is the flow of energy from the large scales to the small scales, and vice versa. The TKE is computed by taking the Fourier transform $\hat{v} = (\hat{v}_0, \hat{v}_1)$ of the velocity field $v = (v_0, v_1)$, and computing the kinetic energy in the Fourier domain

$$\mathrm{TKE} = \frac{1}{2}(\hat{v_0}^2 + \hat{v_1}^2). \tag{21}$$

The energy spectrum is then computed as the sum of the energy content in each 2D wavenumber bucket. Finally, the energy spectrum is averaged over several hundred steps in a long term rollout.

# B    Data generation

For the reference dataset, we use an ensemble of $N$ trajectories, defined on a fine grid $\mathcal{G}$. These DNS trajectories are obtained from a numerical solver using a high order spectral element spatial discretization on the domain $\Omega$, at a temporal resolution of $\Delta t$. These trajectories are then filtered and sampled in time to obtain an ensemble of trajectories $\{\overline{u^{(i)}}\}_{i=1}^N$ defined on the coarse grid $\overline{\mathcal{G}}$ at a temporal resolution of $10\Delta t$. The filtered DNS contains only explicitly the large scales. However, it was obtained from the full order model so it contains the true evolution of the large scales resulting from *the interactions between the large and the small scales*.

We use the spectral element method [23] to obtain trajectories of the N-S equation 2. The solver uses a fine-grained domain $\mathcal{G}$ with $E$ elements and polynomial order $P$ to spatially discretize $\Omega$. These DNS trajectories obtained at fine temporal resolution consistent with the CFL condition. See Appendix C for more details on the numerical solver. We generated trajectories of Kolmogorov flow by running $N = 32$ independent DNS runs over the domain $[0, 1]^2$. We discretized the domain uniformly into $48^2$ elements and solved with a spectral element solver at polynomial order 8, and used a timestep of $\Delta t = 10^{-4}$. For each independent run, we perturbed the initial velocity field with a Gaussian field with mean uniformly in $[0, 1]^2$ and standard deviation of 0.1. We discarded the first $50,000$ steps in order to 'forget' the initialization. The next $128,000$ steps were taken to form the dataset. Then, for the LES we use the same spectral element solver but on a coarse-grained domain $\overline{\mathcal{G}}$ of $12^2$ elements and a polynomial order of 4. The DNS trajectories were filtered on a per-element basis following [12] and interpolated to the coarse grid $\overline{\mathcal{G}}$. We sampled every 10th step to obtain trajectories of 12,800 steps, at a temporal resolution of $10\Delta t = 10^{-3}$. Out of the 32 independent runs, 24 of them were used for training, 4 of them for validation and 4 for the test dataset.

For the flow past cylinder dataset, we used a rectangular domain $[-10, 30] \times [-10, 10]$. The circular cylinder is placed with center at the origin $(0, 0)$ and diameter 1.0. The boundary conditions are periodic along the $y$-axis. The left inflow wall has an inflow of fluid at 1 units per second in the $x$-direction which is implemented via constant Dirichlet boundary condition. The right outflow wall has a homogeneous Neumann condition to allow the fluid to exit the domain. The domain was discretized into 292 quadrilateral elements at polynomial order 13. The temporal resolution of the

DNS simulation was $t = 10^{-4}$. A single long trajectory was generated with $350,000$ timesteps, where the first $150,000$ timesteps were thrown out.

## C  Spectral element Navier-Stokes solver

**Spatial Discretization**   Recall the Navier-Stokes equations Eq. (1).

$$\partial_t u + (u \cdot \nabla)u = -\nabla p + \nu \nabla^2 u, \tag{22}$$
$$\nabla \cdot u = 0. \tag{23}$$

Following the weak formulation of the equations and spatially discretizing via the spectral element method [23]. The velocity field is discretized using a nodal basis at Gauss-Lobatto-Legendre (GLL) points on each quadrilateral element using polynomial order $P$, while the pressure field is discretized using a nodal basis at Gauss-Legendre (GL) points using polynomial order $P - 2$. This staggered discretization is stable and avoids spurious pressure modes in the solution.

The semi-discretized version of the Navier-Stokes equations become:

$$M\frac{du}{dt} + Cu + \nu Ku - D^T p = 0, \tag{24}$$
$$-Du = 0, \tag{25}$$

where $M$ is the diagonal mass matrix, $K$, $D^T$ and $D$ are the discrete Laplacian, gradient and divergence operators, and $C$ represents the action of the nonlinear advection term.

**Time integration**   The time integration is third order, so the values of $u$ and $p$ at the previous three timesteps are used to solve for the new values at the current timestep [23]. A third order backward differentiation formula (BDF3) is used for both linear and nonlinear terms. However, the nonlinear term $Cu$ may require solving a nonlinear implicit relation – to avoid this, the advection term is extrapolated using the third order extrapolation formula (EX3).

**Linear solves**   The one step forward time evolution of the fractional step method involves solving two symmetric positive definite (SPD) linear systems, one for the intermediate velocity which is not divergence free, and next for the pressure correction term. Both the linear solves involve large linear systems, which makes it it infeasible to materialize the dense matrices in memory. Hence, we use a matrix-free conjugate gradients iteration to solve them. Further speedups may be gained by using appropriate preconditioners, which is especially important to overcome the poor conditioning when scaling up to larger systems and higher orders.

**Differentiation**   For the reverse mode differentiation, we simply solve the transposed linear systems during the backward pass. Since the systems are symmetric, we have to solve the same system with a different right hand side. We use Jax's [15] custom automatic differentiation toolbox to override the reverse mode differentiation rule.

## D  Neural SDE solver

The encoder stage reduces the input sequence length $E = 144$ to a a smaller sequence length $E' = 9$; see Appendix G. The neural SDE stage operates on reduced sequence length $E' = 9$, with embedding dimension 192. The drift $h_\Theta$ and diffusion $g_\Theta$ parameterizing the latent Neural SDE uses a Transformer architecture and element-wise multi-layer perceptron (MLP) respectively. The prior drift follows the same architecture as the posterior drift.

We write $\Theta = (\theta, \phi)$ where $\theta$ and $\phi$ are the prior and posterior parameters respectively. The SDE is solved over the unit time interval $\tau \in [0, 1]$ with the state dimension $9 \times 192 = 1728$. The prior and posterior SDEs evolve according to the following equations:

$$d\tilde{Z}_\tau = h_\theta(\tilde{Z}_\tau, \tau)d\tau + g_\theta(\tilde{Z}_\tau, \tau) \circ dW_\tau, \tag{26}$$
$$dZ_\tau = h_\phi(Z_\tau, \tau)d\tau + g_\theta(Z_\tau, \tau) \circ dW_\tau. \tag{27}$$

Note that the diffusion term $g_\theta$ is shared by the prior and posterior SDEs. Both the drift and diffusion functions are also functions of the time $\tau$. Additionally, the posterior drift is a function of an additional context term which is simply fixed at $\mu_{Z_0}$ and does not evolve with time.

Four independent trajectories of both the prior and posterior SDEs are sampled at both training and inference time. The initial state of the prior SDE is sampled from a zero-mean Gaussian of the same dimensionality with constant diagonal variance $0.1^2$. The initial state $Z_0$ of the approximate posterior SDE is obtained from a multivariate Gaussian distribution parameterized by the output of the encoder $\mathcal{E}$. Specifically, the encoder outputs $(\mu_{Z_0}, \sigma_{Z_0})$, and each trajectory of the SDE is sampled independently from a Gaussian with mean $\mu_{Z_0}$ and diagonal variance $\sigma_{Z_0}^2$.

**KL Divergence**   The KL divergence term of the Neural SDE is given by the sum of the KL divergence due to the distribution of the initial value $Z_0$ as well as the entire trajectory $\{Z_\tau\}_{0 \le \tau \le 1}$

$$\mathrm{KL}^{\mathrm{NSDE}} = \mathrm{KL}_{Z_0} + \mathrm{KL}_{\{Z_\tau\}}. \tag{28}$$

The $\mathrm{KL}_{Z_0}$ term is a standard KL divergence between two Gaussians with diagonal covariances, letting $\sigma^{\mathrm{prior}}$ is a fixed hyperparameter set to $0.1$, and resulting in the following:

$$\mathrm{KL}(\mathcal{N}(\mu_{Z_0}, \sigma_{Z_0}) \| \mathcal{N}(\mathbf{0}_n, \sigma^{\mathrm{prior}} \mathbf{1}_n)) = \sum_{i=1}^{n} \log \frac{\sigma^{\mathrm{prior}}}{\sigma_{Z_0}^{(i)}} + \frac{(\sigma_{Z_0}^{(i)})^2 + (\mu_{Z_0}^{(i)})^2}{2(\sigma^{\mathrm{prior}})^2}. \tag{29}$$

The KL divergence term $\mathrm{KL}_{\{Z_\tau\}}$ is given by the following integral

$$\mathrm{KL}_{\{Z_\tau\}} = \int_0^1 \frac{1}{2} \left( \frac{h_\phi(Z_\tau, \tau) - h_\theta(Z_\tau, \tau)}{g_\theta(Z_\tau, \tau)} \right)^2 d\tau, \tag{30}$$

which is computed along with the SDE solve by augmenting the state with a single dimension. The additional scalar KL contribution term is then computed by the forward SDE solve by integrating the drift given by Eq. (30) and zero diffusion.

**Architectural Choices**   The architecture of the drift functions are based on the Transformer architecture. Each Transformer block contains a self-attention and an MLP block, each preceded by a layernorm and GeLU nonlinear activations. Four layers of Transformers were stacked for both the prior and posterior drift. The diffusion functions are parameterized by a non-linear diagonal function, where each output coordinate is a function of only the corresponding input coordinate. Each coordinates diffusion MLP has single dimension input and output with four hidden layers of 32 neurons each. In the diffusion functions tanh activations were used for added stability and the final activation was exponential function so that the output is positive. The total number of parameters in the drift and diffusion functions is 1,862,977.

**Numerical Solver**   The SDE is numerically solved using the reversible Heun scheme [46], which converges at strong order $0.5$ to the Stratonovich SDE solution. A uniform timestep of $0.0625$ is used.

**Backpropagation**   The SDE solver is differentiated through in the optimization process. For the backward pass, an adjoint SDE is solved, following the derivation in [52]. Even though the Itô integral is equivalent to the Stratonovich integral, using the Stratonovich SDE makes the form of the adjoint SDE convenient to derive and constructs a more efficient backward process [46].

## E   Hyperparameters

The following hyperparameters were tuned over the validation set: learning rate, KL penalty and the number of layers in each transformer block. We selected the best performing model based on mean squared error on the validation trajectories, averaged over 8 rollout steps.

**Learning rate schedule**   We trained the model for 25 epochs. The learning rate schedule followed a linear warmup phase from $0$ to the base learning rate $\alpha$ over the first epoch. Over the remaining 24 epochs, the learning rate decays to $0$ according to cosine decay schedule. The base learning rate $\alpha$ is tuned separately for both the niLES and the deterministic NN model from among a logarithmic distribution of learning rates.

**KL penalty** The KL penalty term follows a linear warmup to the final value $\beta$ over the first ten epochs. Beyond that, the KL penalty remains fixed at the same value for the rest of the training period. The KL penalty term $\beta$ was selected from among the values 0.001, 0.01, 0.1, 1. and 10.

**Number of layers** The number of layers in each phase of the MViT (see Appendix G) is tuned. The validation error is found to saturate at 6 layers for each phase, while the inference time degrades with increase in the number of layers.

Both deterministic NN and the niLES model were hyperparameter-tuned in the same way, except when the hyperparameter (such as KL penalty) were not applicable to the deterministic NN model.

# F Computational resources

We used 8 Nvidia V100s to train both the deterministic NN baseline and the niLES model. Training was done for 25 epochs of the training data which took up to 20 hours. For the inference phase we used Google Colab runtime with a single Nvidia V100. For the spectral element numerical solver, we used double (float64) precision. The model parameters and intermediate model variables during training and inference used single (float32) precision.

# G Encoder-decoder architecture

The overall architecture of both the niLES model and the deterministic NN model consists of a multiscale ViT (MViT) structure [27, 53]. The architecture of the deterministic NN consists of an encoder and decoder, where the encoder follows the MViT architecture. The encoder maps the input to a much lower latent dimension, while decoder mirrors the same architecture in the reverse sequence. The total number of parameters in the Encoder-Decoder is 2,752,178.

**Encoder** The encoder $\mathcal{E}$ takes as input the LES field $v$ defined on the coarse-grained discretization $\overline{\mathcal{G}}$. We tackle 2D geometries using a mesh of $E$ elements, each possessing $(P+1)^2$ nodal points at which the velocity is defined where $P$ is the polynomial order. The field $v$ may be interpreted as seqeuence of $E = 144$ tokens, with an input embedding dimension of $d(P+1)^2$. In the first layer, the input is projected to an embedding dimension $W = 48$.

Taking cues from MViT [27, 53]; in a sequence of stages we decrease the number of tokens from $E$ to $E' = 9$ ($E' \ll E$), and each stage has $M = 6$ layers of Transformer blocks. In each stage, the token length is reduced by 4x and the embedding dimension is doubled. The reduction of the number of tokens by attained by mean-pooling the embeddings of the constituent tokens. The increase in the embedding dimension is facilitated by the last MLP in the transformer block. We have a total of two stages, so the final token length is therefore $144 \div (4 \times 4) = 16$ and the final embedding dimension is $48 \times 2 \times 2 = 192$.

**Decoder** The decoder $\mathcal{D}$ is similar to the encoder, except it operates in reverse order: i.e., it increases the sequence length from $E'$ back to $E$. The same number of stages are employed, with $M$ layers of transformer blocks in each stage. It has the same number of parameters as $\mathcal{E}$. The decoder, mirroring the encoder, increases the token length by a factor of four at the end of each stage, and decreases the embedding dimension by two.

**Skip Connections** At the end of every stage of the MViT, a skip connection is added between corresponding layers from the encoder or decoder. For instance, there is a skip connection from the end of the last encoder stage to the beginning of the first decoder stage.