# OpenReview forum: "Neural Ideal Large Eddy Simulation: Modeling Turbulence with Neural Stochastic Differential Equations"
_NeurIPS.cc/2023/Conference — NeurIPS 2023 poster_

### Official Review · Reviewer_hr9G · 2023-07-02

**Soundness:** 3 good
**Presentation:** 4 excellent
**Contribution:** 4 excellent
**Rating:** 8
**Confidence:** 5

**Summary:**

The authors introduce a neural SDE model for LES flow fields. The model structure is motivated by the ideal LES approach and the model learns a data-driven closure term. The learned latent representation captures the variability and fine-scale structure of the fully-resolved DNS solution that is lost by the LES filter.

**Strengths:**

The paper is excellently motivated, presented and embedded into the theory. Stochastic modeling for turbulence as presented in this paper is an important direction to explore and therefore the paper would be valuable contribution to the field.

**Weaknesses:**

1. The authors seem to use the term "closure model" more loosely than usual in the introduction. Out of the citations for data-driven closure models in line 39, only [45] defines a closure model as the term is used in the context of RANS and LES.
1. Lines 177-182 are slightly misleading as they can be understood to suggest that the model would learn possible DNS realizations as $Z_t$ instead of an opaque latent representation. (Actually, lines 211-213 clarify this but could be moved forward/merged into the first paragraph)
1. A figure showing the flow (and predictions) after, for example, 200, 400, etc. steps would help the reader appreciate how chaotic the flow is and how much the velocity field evolves over 800 steps.
1. A figure comparing the DNS and filtered data would also be helpful for the reader to understand the effect of LES filtering better.

**Questions:**

1. Eq (2): Why can the pressure be eliminated?
1. Line 87: Which boundary conditions do you impose and how do you do so implicitly?
1. Is there a mix-up of $h_\theta$, $h_\psi$, $h_phi$, $g_\theta$ and $g_\phi$ happening in Eq (8), (9), (10)?
1. Eq (14): You could define $f$ more explicitly here.
1. Eq (16): What do you mean with spatio-temporal lifting? Does w correspond to z? What is $\mathcal{D}$?
1. What exactly is the relationship between the SDE solver timestep and the simulation timestep? Are the temporal intervals $\mathcal{T}_i$ rescaled to $[0, 1]$?

**Limitations:**

The authors discuss limitations adequately.

---

> ### Author Rebuttal · Authors · 2023-08-09
>
> Thank you for the detailed and thoughtful review. We are glad that you found our paper well-motivated and that it provides an important direction to explore, viz. stochastic modeling of turbulence using neural networks.
>
> **Writing**
>
> >The authors seem to use the term "closure model" more loosely than usual in the introduction. Out of the citations for data-driven closure models in line 39, only [45] defines a closure model as the term is used in the context of RANS and LES.
>
> That is a good catch! We’ve indeed used broader strokes in the introduction to give an overview of the extensive literature in closure models and other forms of data-driven approaches for turbulence. We will fine-tune the writing in the introduction to reflect the specific nature of closure modeling for LES and RANS that we undertake in this work.
>
> >Lines 177-182 are slightly misleading as they can be understood to suggest that the model would learn possible DNS realizations as Z_t instead of an opaque latent representation. (Actually, lines 211-213 clarify this but could be moved forward/merged into the first paragraph)
>
> In line 177-182, we started the explanation by introducing one idea at a time, that of the need to model the DNS, and later on specify we just need a latent representation and not the whole DNS.  You are right that it can create confusion as to what our method is actually doing. Instead of saying that we “generate DNS trajectories” we can be more explicit at the beginning of the first paragraph that we learn a representation of said DNS trajectories. Furthermore we can clarify in these paragraphs that we indeed learn an opaque representation in latent space for these DNS trajectories.
>
> **Figure showing the flow evolution and the DNS snapshots with corresponding filtered versions**
>
> Thank you for the excellent suggestions! We will include these figures in the camera-ready version of the paper.
>
> **Elimination of pressure**
>
> The pressure can be eliminated in the Navier Stokes equations by e.g. taking the divergence of the momentum equation. We use the fact that the divergence of velocity is zero. It leads to a Poisson-like equation for the pressure, which can then be substituted into the momentum equation again. This results in an equation where the velocity is the only unknown. For more details, we provide the reference [1, page 295, section 6.2.2]. We will also add the reference to the corresponding sentence in the camera-ready version of the paper.
>
>
> **Boundary conditions**
>
> The boundary conditions we impose in the Kolmogorov flow are 2D periodic along both axes. In the newly added cylinder wake dataset, we follow a setup of a static circular cylinder with no slip boundary condition on the cylinder surface, constant Dirichlet boundary condition on the inflow wall, periodic along the vertical axes, and homogeneous Neumann on the outflow wall. (Please see our response to reviewer SnNJ for more details.)
>
> In line 87, by ‘implicit’ we meant that the boundary conditions are assumed to have been enforced in subsequent equations (Eq. 4, Eq. 5) without mentioning the fact explicitly. We can see how it may cause confusion with implicit numerical methods or linear solves. We apologize for the oversight. We shall rephrase this section to avoid confusion.
>
> **Notational issues in Eq (8), (9), (10) and (14)**
>
> Thank you for catching the error. Eq (8) should say $g_\theta$ instead of $g_\phi$; and Eq (9) should say $h_\phi$ instead of $h_\psi$. We will fix these in the manuscript. Appendix D also reiterates the neural SDE equations with the corrected notation.
>
> Thank you for the suggestion on Eq (14), in the text we can spell out $f$ in terms of the small scale fluctuations induced on the LES field from the DNS field.
>
> **Spatio-temporal lifting and SDE solver/simulation timesteps**
>
> We refer to the DNS trajectory $w$ as being the spatio-temporally lifted version of the latent space trajectory $Z_t$, as the $w$ lives in both spatially higher resolution as well as temporally in the finer timescale. $\mathcal{D}$ is the decoder (defined on line 203) which sends the latent space trajectory $Z_t$ to the fluctuation on the LES field caused by the DNS trajectory $w$. The output space of $\mathcal{D}$ has the same dimensionality as the LES field.
>
> The SDE timesteps are the same as the DNS simulation timesteps, which are an order of magnitude smaller than the LES simulation timesteps. They are not same as $\mathcal{T}_i$ (or rather the timesteps we use for the filtered DNS training data) since the filtered DNS trajectories are downsampled by ~10x. As the SDE is meant to capture the fluctuations in the fast timescale, in each LES step the SDE evolves forward by ~10 timesteps. We do rescale the timesteps to $[0, 1]$ but that is for simplicity and is an implementation detail.
>
>
>
> **References**
>
> 1. Deville, M. O., Fischer, P. F., & Mund, E. H. (2002). High-order methods for incompressible fluid flow (Vol. 9). Cambridge university press.

---

> > ### Comment · Reviewer_hr9G · 2023-08-13
> >
> > Thank you for the clarifications. I will remain with my assessment.

---

### Official Review · Reviewer_a9bB · 2023-07-07

**Soundness:** 3 good
**Presentation:** 3 good
**Contribution:** 3 good
**Rating:** 6
**Confidence:** 3

**Summary:**

This paper introduces a data-driven method for approximating the closure term in Large Eddy Simulation (LES). The closure term represents the unresolved scaling effect caused by reducing the computational grid through downsampling. In comparison to the traditional physically-informed approach, the learned closure term has the ability to automatically capture relevant features without the need for specific domain expertise or manual design. The model improves temporal resolution by employing latent space evolution with stochastic propagation. The proposed model demonstrates competitive performance compared to the Filtered DNS reference across different wave numbers, and significantly outperforms the baseline model.

**Strengths:**

- The proposed method outperforms several baselines in terms of the squared error and the accuracy of turbulent kinetic energy spectrum.
- The latent space evolution implements the procedure to solve the SDE equation, which introduces the domain specific inductive bias into the model.

**Weaknesses:**

- The model's training solely relies on a single reconstruction loss, which is atypical for stochastic latent variables. The absence of prior regularization with respect to latent space implies that the stochastic component added at each temporal step acts more like a sparsity regularization, as the model is trained to minimize the stochastic effect rather than explicitly capturing the physical dynamics. Apart from prediction accuracy, there is a lack of additional benchmarks to demonstrate the advantages of the stochasticity, such as prediction diversity.

- When comparing the proposed niLES with deterministic Neural Network-based models, it should be noted that niLES introduces additional parameters $h_{\phi}$ and $g_{\theta}$ as well as more computational routes, potentially leading to an unfair comparison. The authors should showcase the model's performance while varying the magnitude of the Wiener process in Equation 15 and compare it to the scenario where the noise magnitude is zero.

- The current evaluation protocol does not adequately assess the scalability of the model, particularly with respect to wave numbers. To address this limitation, the authors should exclude a certain range of wave numbers as test cases, enabling a more comprehensive evaluation of the model's scalability.

**Questions:**

Q: It would be better if the authors could show the stochastic property in latent space brings diverse and valid results in the LES space.

---

> ### Author Rebuttal · Authors · 2023-08-09
>
> Thank you for your detailed review. Please find our response below.
>
> **Single reconstruction loss**
>
> We apologize for the confusion. Eq. (19) indeed contains only the reconstruction loss, which is only a fraction of the training loss we ultimately use for training (see line 218 right after Eq(18)). Eq. (30) in Appendix D shows other loss terms related to the latent space. We will refactor this section and add these additional terms to the main text.
>
> **Prediction diversity**
>
> This is a great suggestion. We have added a plot in the uploaded PDF (Fig 3) to show how the rollouts in the LES space vary due to the stochastic variation of the latent space trajectories.
>
> **Magnitude of the Wiener process**
>
> We are adding a new baseline where the magnitude of the Wiener process is zero; see Figs 1,2,5 in the attached PDF. This corresponds to a deterministic NN whose latent space dynamics are parameterized as a neural ODE. We added additional numbers of layers (3x more layers) in the neural ODE drift function to compensate for the fact that the neural SDE contains additional parameters corresponding to prior drift and diffusion. The resulting neural ODE-based architecture has strictly more parameters than our proposed model niLES.
>
> **Scalability of the model**
>
> Thank you for your feedback on increasing the comprehensiveness of the evaluation. However, we are not fully sure that we understood what you mean by “exclude a certain range of wave numbers”. Could you please clarify your comment?
>
> To broaden our evaluation, we have added several more test cases for the evaluation of our model. In particular, for the Kolmogorov dataset we have included the RMSE and TKE spectrum for 6 independent examples. (See Figs 1 and 2 in the attached PDF.)
>
> Additionally, we have included another scenario – cylinder wake at Reynolds number 500 (some more details are in our response to reviewer SnNJ). This scenario has more than 5x the degrees of freedom (52K vs 9.2K) in the DNS simulation compared to the Kolmogorov flow and exhibits irregular, chaotic vortex shedding. The LES simulation in the cylinder wake case contains 7.7K degrees of freedom compared to 2.3K in the Kolmogorov LES field.
>
> For the cylinder wake dataset, we have included the average RMSE and the spread (min / max) as obtained by our method and the baseline methods. See Fig 5 in the attached PDF. We hope this expands the evaluation satisfactorily.

---

> > ### Comment · Reviewer_a9bB · 2023-08-20
> > **Further questions**
> >
> > I appreciate the authors’ efforts to address my inquiries and share results related to neural ODE and the description of sample diversity.
> > Having gone through the feedback provided by the authors, I have further questions regarding the results, particularly on the comparison to deterministic methods.
> >
> > (1) Concerning the training objectives, the introduced approach learns features in the latent space through VAE-like objectives. Yet, it outperforms deterministic methods when considering reconstruction error. This seems somewhat unexpected to me, given that a standard VAE often yields blurred results and struggles to retain high-quality details.
> >
> > (2) Referring to Figure 4, the deterministic NN (d) seems to generate a sharp output, but it concurrently leads to high-frequency artifacts. My impression is that these artifacts might be attributed more to specific model configurations and training issues rather than an inherent property of deterministic NNs.
> >
> > (3) When it comes to assessing the diversity of samples, how can one accurately evaluate their correctness beyond mere qualitative visualization?

---

> > > ### Author Response · Authors · 2023-08-20
> > >
> > > Thank you for the positive comments regarding our responses and your thoughtful follow up questions. Please find our replies below, and let us know if you have any further questions.
> > >
> > > (1) We compare the reconstruction errors from long-term rollouts. For short-term rollouts (i.e., when the model is unrolled only 8 times, and that is identical to the setup during training) the deterministic approaches and our method yield similar reconstruction errors.
> > >
> > > In a nutshell, the closure model learns to correct the coarse solver. You have noted correctly that VAE-like objectives have been shown to have a ‘smoothing’ effect on the predictions, which also aids in generalization. However, in our method, the LES samples are not directly samples from the VAE, but rather the VAE-like samples form the fluctuations or corrections to the LES solver. Therefore, the smoothing effect of a VAE does not necessarily translate to the smoothing in the LES field samples (because we do not use those samples as the final output).
> > >
> > > In fact, we believe that the smoothing-like effect in the fluctuation-space, which dampens the overcorrection of deterministic methods, is crucial to preventing the unphysical energy buildup. This latter energy buildup prevents the highly chaotic system from retaining high-quality details and leads to loss of accuracy over the long term.
> > >
> > > (2) As you note correctly, the deterministic NN approaches maintain a sharp output over the short term, but over the long term this leads to instabilities and unphysical artifacts. The high-frequency artifacts are caused due to the unphysical build up of turbulent energy over time at the higher frequencies (see Fig 3 in the main paper). An extremely well-tuned deterministic NN might be able to attenuate those issues but such unphysical buildup would eventually lead to the simulation ‘blowing up’. This phenomenon is well known in computational fluid dynamics, and it is usually handled by incorporating a filtering stage that removes spurious high-frequency energy at each time-step (as is done in the Implicit LES). Such filtering is based on the spectral properties of the time-stepper, and it is tailored to remove only the spurious components, thus maintaining the high-accuracy of the solver in a stable manner. In an abstract level, we can argue that in our formalism the averaging of the fluctuations is analogous to such filtering.
> > >
> > > The inability to retain high-quality physical features precisely highlights the issues of treating the learning signal as a deterministic LES field. In other words, we view the better performance largely due to the probabilistic formalism of learning LES, which is now tractable through neural-SDE based modeling.
> > >
> > > (3) You have raised a good point: one possibility is to compute a large number of ‘nearby’ DNS trajectories, and filter them to LES fields. We can then use our samples to compare the generated distribution to the ensemble of the LES fields. While this is theoretically plausible, we need to overcome the challenge of computing nearby DNS trajectories and comparing high-dimensional distributions, which is computationally expensive. We agree that these comparisons would be fruitful directions to explore in future work.

---

> > > > ### Comment · Reviewer_a9bB · 2023-08-21
> > > > **(Updated) Thanks for your clarification**
> > > >
> > > > Thank you for your clarification. The discussion related to the short-term and long-term rollouts performance will be helpful for better accessing the method. I will update my score to 6.

---

> > > > > ### Author Response · Authors · 2023-08-21
> > > > >
> > > > > Thank you for the positive feedback. We will update the final version with the discussion regarding the accuracy of short-term and long-term rollouts.

---

### Official Review · Reviewer_GBgQ · 2023-07-07

**Soundness:** 3 good
**Presentation:** 3 good
**Contribution:** 3 good
**Rating:** 6
**Confidence:** 3

**Summary:**

This submission proposed a data-driven method to learn a.closure model to simulate the results from DNS. The key part is a latent stochastic process by Neural SDE. And finally compute the Monte-Carlo approximation.

**Strengths:**

1. The model treats the DNS as a stochastic process, instead of a deterministic process as in many previous works.

2. Empirical results indicate it performs well on Kolmogorov flow.

**Weaknesses:**

1. The number of datasets is small.
2. It lacks recent deep learning-based methods as baselines.

**Questions:**

1. Since the work treats DNS field as stochastic, why can you compute RMSE in the figure 3?
2. Why do you design a stochastic process on latent space, instead of directly on original field?

**Limitations:**

See questions and weakness.

---

> ### Author Rebuttal · Authors · 2023-08-09
>
> Thank you for your review and feedback.
>
> **RMSE in Fig 3**
>
> You are correct that we treat DNS fields stochastically. However, the ideal LES field is deterministic, which is approximated by a filtered DNS instance, and is a valid approximation for short time rollouts.
>
> Over longer time horizons, however, chaoticity dominates. In that regime RMSE ceases to be a meaningful metric; here we have provided statistical error metrics such as Turbulent Kinetic Energy (TKE) spectrum.
>
> **Baselines and number of datasets**
>
> We have added cylinder wake as an additional dataset showcasing the performance of our method. In addition, we added another deterministic NN-based LES as baseline using the Neural-ODE framework. Please see our response to reviewer SnNJ. The conceptual framing of using SDE for probabilistic LES is the first kind to our best knowledge, so we do not have such probabilistic NN-based baselines to directly compare to. We hope our method becomes a baseline for future work.
>
> **Why not latent space on the original field?**
>
> The main objective of LES modeling is the reduction of the computational cost. In general, the LES formulations tend to be more expensive at the same resolution than the original systems. However, the cost reduction is achieved by using a much coarser grid compared to the original one, which, compounded by the larger time-steps, results in an overall cheaper method.
>
> In our case, running an SDE at the original resolution would be equivalent to running the DNS, which would ultimately defeat the purpose of LES.
>
> The main insight/inductive bias we would like to incorporate is that: instead of running multiple DNS trajectories, we assume that there is a low-dimensional latent space where we can efficiently sample multiple short-term trajectories; and whose aggregated statistics remain close to the statistics stemming from directly computed DNS trajectories.

---

> > ### Comment · Reviewer_GBgQ · 2023-08-20
> >
> > Thanks for the authors' reply and new results. I think it solves my concerns well. I will keep my score.

---

### Official Review · Reviewer_SnNJ · 2023-07-18

**Soundness:** 3 good
**Presentation:** 3 good
**Contribution:** 2 fair
**Rating:** 7
**Confidence:** 4

**Summary:**

This paper targets learning turbulence closure models for RANS simulations via neural networks. The paper proposes to use a neural SDE on the latent space of a transformer to predict different samples from the distribution of the next state, and then compute an average over these. This process is unrolled and trained for a sequence of multiple steps.

**Strengths:**

Overall, this is a good idea, and I'm not aware of a an NSDE being previously used in this form. Thus, I see the general direction of the paper and the promising approach as strong points.

**Weaknesses:**

On the other hand, the paper targets a single, two dimensional Kolmogorov flow secnario as the only test case. In addition, only a single deterministic NN is compared to (plus an implicit LES solver). For this single data set, the paper is lacking a stable evaluation: multiple, differentily initialized models evaluated across multiple tests to obtain a stable result are not evaluated. In addition, NeurIPS is a very broad ML venue. Turbulence is definitively an exciting topic here, but even more important would be a broader evaluation, ideally with substantially different secnarios to show that the method has merit beyond turbulence. In its current form, I don't think that the results are sufficient for a NeurIPS paper.

I see two ways to improve this aspect of the submission: either the authors focus their writing on the turbulence secnario (cf. below), and present multiple scenarios in this context, or non-turbulence cases are included to broaden the scope. A more stable evaluation with multiple models and tests should be included either way. This potentially also could help to show the benefits of the method more clearly. Right now the gains in terms of accuracy and the differences in the TKE spectrum seem to be mild. Additional cases could show areas where the approach gives larger improvements.

In addition, I would also recommend that the authors include additional learned baselines. This is less crucial, but would nonetheless help to put the work into the context of previous methods at NeurIPS, ICLR & co.

I also do want to mention two weak points in the writing. One is that I found the motivation (esp. L44) quite unintuitive: the "inductive bias of the LES field" is not very clear, and the summary implies this is "simply" a matter of choosing the right architecture. Things get clearer afterwards, but reading the paper front to back, I think this summarizing question is not helping a reader.

The conclusions are also not a good fit with the rest of the paper: suddently, a transition is is made to generic chaotic systems. The whole previous paper targets a single specific scenario in the form of Kolmogorov turbulence, and presents a single set of results. Hence, this outlook is not supported by the content of the paper. I can understand that the authors have hopes that their method will at some point in the future generalize to other cases, but it should be made clear that this is an outlook. Rather, references to specific works where the authors see potential would be interesting to give here.

Overall, I want to encourage the authors to continue their direction of work. Nonetheless, I find it difficult to directly argue for accepting this paper in its current form.

**Questions:**

Which other, specific applications and scenarios do the authors see for their method?

**Limitations:**

Limitations are discussed briefly.

---

> ### Author Rebuttal · Authors · 2023-08-09
>
> Thank you for your thoughtful review and detailed feedback. We appreciate your encouragement and your suggestions on how to improve the quality of the manuscript.
>
> **More scenarios**
>
> We have included in the uploaded PDF an instance of our proposed methodology applied to cylinder wake at a high Reynolds number of 500, where the system exhibits irregular vortex shedding and chaotic flow. The difficulty of such an example is two-fold [3]: the need of a nonuniform mesh that is usually refined near the cylinder, and the need to capture the boundary layer near the cylinder. We hope this adds diversity to the turbulent flows that this method could be applied to. See Fig 4 in the uploaded PDF for a qualitative rollout and Fig 5 for the accuracy plots with the new dataset. We have also shown the spread (min/max values) among 10 testcases in the dataset.
>
> **Stable Evaluation**
>
> Thank you for the suggestion. For both the Kolmogorov datasets, we have added evaluations across 6 different test cases (See Fig 1 and Fig 2 in the attached PDF) We plot both the RMSE and the TKE among the 6 test cases. While our training method unrolls for only 8 steps, the evaluation is unrolled to 100s of steps, well beyond the training horizon.
>
> **Why turbulent flows?**
>
> > I see two ways to improve this aspect of the submission: either the authors focus their writing on the turbulence secnario (cf. below), and present multiple scenarios in this context, or non-turbulence cases are included to broaden the scope
>
> Thank you for the comment. We target turbulent flows as a prototypical chaotic system with an impact in real world applications encompassing engineering and science, such as weather and climate.  In fact, many of the difficulties inherent to chaotic systems manifest in turbulent flows. For example, maintaining long-term statistics accurately (such as turbulent kinetic energy) in turbulent flows is especially challenging. Many data-driven approaches become unstable for longer rollouts [1,2]. Your suggestion of investigating the method’s applicability to other flows is appreciated and as mentioned above we have added a new setup for such flows.
>
> We will modify the text in order to convey this point more clearly.
>
> **Gains in Accuracy and TKE**
>
> While the gains in accuracy shown in Fig. 3 (of the manuscript) are seemingly modest, achieving similar gains using traditional computational techniques used in turbulent flows would require the computation of an expensive direct numerical simulation (DNS). For instance, using a high-order DNS that achieves a similar quality is roughly two orders of magnitude slower.
>
> To the best of our knowledge, similar works on Neural Network-based turbulence closure models do not achieve such gains at such a high Reynolds number (20,000) while being stable for long beyond the training horizon. Furthermore, among spectral element methods, the Implicit LES method is considered to be the state-of-the-art [4]. Even against this approach we have demonstrated that our method produces more accurate simulations while exhibiting long-term stability.
>
> **Additional baselines**
>
> Thank you for the suggestion. We are including a Neural ODE model [5] as a learned baseline for deterministic LES.
>
> As you pointed out, one of the novel aspects of our approach is leveraging the probabilistic formalism of LES to design the algorithmic pipeline of our method. To the best of our knowledge, this has not been proposed before, therefore, we were not able to find other probabilistic LES  approaches to be used as learned baselines.
>
> **Writing**
>
> We acknowledge your feedback on the clarity of the writing.
> We will properly nuance the outlook section of the paper, particularly the connection of the results shown in this manuscript and the broader area of chaotic dynamics. We will also shift some of the explanations through the paper to better articulate how the motivation and modeling insights guided our choices in the algorithmic pipeline, in particular the choice of an NSDE in latent space to simulate DNS efficiently as a closure model.
>
> **References**
>
> 1. Beck, A., & Kurz, M. (2021). A perspective on machine learning methods in turbulence modeling. GAMM‐Mitteilungen, 44(1), e202100002.
> 1. Moser, R. D., Haering, S. W., & Yalla, G. R. (2021). Statistical properties of subgrid-scale turbulence models. Annual Review of Fluid Mechanics, 53, 255-286.
> 1. Williamson, C. H. (1995). Vortex dynamics in the wake of a cylinder. In Fluid vortices (pp. 155-234). Dordrecht: Springer Netherlands.
> Bosshard, C., Deville, M. O., Dehbi, A., & Leriche, E. (2015).
> 1. UDNS or LES, that is the question. Open Journal of Fluid Dynamics, 5(04), 339.
> 1. Chen, R. T., Rubanova, Y., Bettencourt, J., & Duvenaud, D. K. (2018). Neural ordinary differential equations. Advances in neural information processing systems, 31.

---

> > ### Comment · Reviewer_SnNJ · 2023-08-19
> > **Rebuttal**
> >
> > I’d like to thank the authors for the comments and the updated results.
> >
> > The cylinder case is very good to see! In terms of evaluation, I would encourage the authors to provide an averaged evaluation across multiple trained models (with different random seeds) for a final version.
> >
> > Nonetheless, I’d be happy to support an accept for this paper, and I’ve raised my score.

---

> > > ### Author Response · Authors · 2023-08-21
> > >
> > > Thank you for the positive feedback. We will update the final version with the new dataset and expand the evaluation with multiple trained models initialized with random seeds.

---

### Author Rebuttal · Authors · 2023-08-09

We thank all reviewers for providing thoughtful reviews and constructive feedback. We are encouraged by the positive comments that our method is well-motivated, clearly presented and provides an important direction to explore in the area of data-driven turbulence closure modeling.

To address the reviewers’ concerns regarding the need for a broader evaluation, we have made the following high level changes:

- **Additional learned baseline.** We have added an additional deterministic NN-based baseline method: this architecture uses the same encoder-decoder but uses a Neural ODE based latent evolution. This corresponds to zeroing out the stochasticity due to the Wiener process in the Neural SDE formulation. Furthermore, to address the concern that the number of parameters in the Neural SDE might be higher because of the prior, posterior drift and diffusion, we have used 3x more layers (12 vs 4) in the drift function of the Neural ODE to compensate for this effect. We have reevaluated the test cases for Kolmogorov flow in Figs 1 and 2 of the attached PDF including the new baseline, and also included four additional testcases from the Kolmogorov flow dataset.
- **Additional turbulence scenario.** We have added an additional dataset – cylinder wake, which is a challenging instance of a chaotic Navier Stokes flow. In addition to its complex geometry, this flow has a high Reynolds number of 500, which is known to exhibit irregular vortex shedding. We have added the evaluation pipeline for our method and the baselines using this new dataset. Our response to reviewer SnNJ contains more details. See Fig 4 in the attached PDF for a qualitative plot and Fig 5 for accuracy plots.

Furthermore, we have addressed (or will address in a final version) the comments regarding typos and clarity issues pointed out by the reviewers. We hope that our explanations are satisfactory, and we are happy to answer any further questions that the reviewers may have.

---

### Decision · Program_Chairs · 2023-09-21

**Decision:**

Accept (poster)

**Comment:**

This paper proposes a new methodology for designing closure models that utilizes a neural SDE incorporating the inductive bias of ideal LES. A Transformer based architecture is employed for the encoder-decoder mapping between the LES field and a latent space. The numerical experiments properly show the effectiveness of the proposed method niLES.
The idea of the proposed algorithm is interesting and looks promising. The stochastic modeling would be a meaningful direction that should be explored by follow-up researches. In summary, I think this paper deserves publication in NeurIPS2023.